# Fresh keeping decision and coordination of fresh agricultural products supply chain under carbon cap-and-trade

**Yang Yang**[1], **Guanxin Yao**[1,2]*

1 School of Management, Jiangsu University, Zhenjiang, PR China, 2 Jiangsu Modern Logistics Research Base, Yangzhou University, Yangzhou, PR China

* gxyao@yzu.edu.cn

## Abstract

Considering the carbon emissions caused by the fresh-keeping of fresh agricultural products, this paper studied the rules of fresh-keeping decision-making in the two-echelon fresh agricultural product supply chain led by suppliers under the carbon cap-and-trade policy. In addition, we designed two contracts, cost-sharing contract and two-part pricing contract, to coordinate the supplier's fresh-keeping decision and the supply chain's revenue. The results show that: whether the carbon cap-and-trade policy is implemented or not, the higher the consumer's preference for freshness and the lower the price sensitivity coefficient of consumers, the more favorable it is for suppliers to improve their fresh-keeping efforts. Under the implementation of carbon cap-and-trade policy, suppliers' fresh-keeping efforts are only related to the carbon transaction price, not the carbon cap; the higher the carbon transaction price, the less the supplier's fresh-keeping efforts, but the more the supplier's income; the smaller the cost coefficient of carbon emission reduction and the larger the coefficient of carbon emission reduction, the more favorable it is for suppliers to increase their fresh-keeping efforts; both cost-sharing contract and two-part pricing contract can coordinate the supply chain of fresh agricultural products, but they have different application scope and coordination effect. These conclusions are of great significance to the operation and management of fresh agricultural products suppliers, the improvement of consumers' quality of life and the protection of ecological environment under carbon cap-and-trade.

## 1 Introduction

In recent years, consumers have higher and higher requirements for the freshness of fresh agricultural products, which drives suppliers of fresh agricultural products to use low-temperature and constant-temperature preservation equipment more frequently to improve the freshness of fresh agricultural products. However, the use of fresh-keeping equipment will increase the consumption of electricity and refrigerant, and bring a lot of carbon emissions. At the same time, on September 22, 2020, at the 75th UN General Assembly, the Chinese government clearly put forward the emission reduction target of "carbon dioxide emissions should reach the peak by 2030 and strive to achieve carbon neutrality by 2060", ang implemented a number

**Data Availability Statement:** All relevant data are within the paper.

**Funding:** The authors gratefully acknowledge financial support from National Natural Science Foundation of China (Grant No.72103178), Social Science Foundation of Jiangsu Province (Grant

No.20GLA002), Graduate Research and Innovation Projects of Jiangsu Province (Grant No. SJKY19_2519).

**Competing interests:** The authors have declared that no competing interests exist.

of carbon limit trading pilots. These pilots include a number of fresh agricultural products suppliers, such as Shanghai Qingmei Green Food Co., Ltd., Bright Dairy Co., Ltd., Duoyu Food (Shenzhen) Co., Ltd., Shenzhen Agriculture, Animal Husbandry and Meiyi Meat Industry Co., Ltd. and so on. On February 1st, 2021, the Chinese government issued the Measures for the Administration of Carbon Emissions Trading (Trial), which extended the implementation of carbon cap-and-trade to the whole country from the previous eight pilot areas, which indicated that the national carbon emissions trading market had been fully launched. Carbon cap-and-trade means that the government decomposes the total amount of carbon emission rights into a certain unit of carbon emission rights, and then allocates the emission rights to carbon dioxide emission source enterprises in a specific way, and allows them to buy and sell carbon emission rights in the market. Fresh-keeping is not only an important part of the operation of fresh agricultural products supply enterprises, but also the main source of carbon emissions of fresh agricultural products supply enterprises. Therefore, the full implementation of the carbon cap-and-trade policy has brought a difficult problem to the suppliers of fresh agricultural products: improving fresh-keeping efforts will bring more carbon emission costs, while reducing fresh-keeping efforts can reduce carbon emission costs but cannot meet consumers' demand for freshness.

Then, under the implementation of carbon cap-and-trade policy, how should suppliers of fresh agricultural products make fresh-keeping decisions and maximize the benefits of the supply chain? This is the main problem of this study. Based on this, we further put forward two questions: ① What impact does the implementation of carbon cap-and-trade policy have on the fresh-keeping decision of suppliers of fresh agricultural products and the benefits of supply chain? ② Under the carbon cap-and-trade policy, how should we design the contract to realize the Pareto improvement of fresh-keeping efforts and supply chain revenue?

The fresh-keeping decision and coordination of fresh agricultural products have always been the focus of scholars [1–4]. Some scholars have discovered the influencing factors of fresh-keeping decision-making of fresh agricultural products enterprises. Liu et al. [5] studied the pricing and fresh-keeping decision-making of dual-channel supply chain of fresh agricultural products under the uncertainty of demand information, and found that inventory and shortage cost would reduce the manufacturer's optimal pricing and fresh-keeping efforts; Bai et al. [6] established a fresh e-commerce supply chain model composed of an online retailer and a third-party logistics provider, and found that the basic quantity loss rate and freshness sensitivity can improve the fresh-keeping work of the third-party logistics provider; Hu et al. [7] found that fairness has a certain impact on retailers' investment in preservation; Yang and Chen [8] studied the pricing and fresh-keeping strategies of suppliers and retailers in different operation modes under the condition of considering consumers' channel preferences and retailers' investment in fresh-keeping; Zhang et al. [9] found that consumers' higher freshness sensitivity is beneficial to the application of preservation technology; Yan et al. [10] explored the impact of fairness concerns on decision-making in the supply chain of fresh agricultural products, and found that manufacturers' behavioral tendencies out of fairness considerations would reduce fresh-keeping efforts. Some scholars have studied the contract design considering the preservation level. Zhang and An [11] constructed a dual-channel supply chain model of fresh food considering the fair concern behavior of retailers, and put forward a revenue sharing contract to coordinate the supply chain; Liu et al. [12] analyzed the influence of freshness demand elasticity and service demand elasticity on the optimal decision under the assumption that both preservation efforts and service level will affect the market demand of fresh agricultural products, and then designed a contract to coordinate the supply chain; Cao et al. [13] took the supplier-led dual-channel supply chain as the research object, and designed three contracts to coordinate the supply chain considering the influence of freshness on

demand and preservation efforts; Luo *et al.* [14] constructed the freshness dynamic game model based on temperature control input and the demand function of fresh agricultural products through the differential game method, and designed a coordination mechanism of "mutual cost-sharing contract with fixed subsidies" to optimize the temperature control investment level of retailers, suppliers and third-party logistics.

In addition, with the implementation of the carbon emission cap-and-trade policy, scholars have also noticed the impact of low-carbon development on the fresh-keeping decision-making of fresh agricultural products enterprises. Bai *et al.* [15] studied the coordination and optimization of the secondary supply chain of fresh agricultural products under the carbon quota and trading policy when the deterioration rate of fresh products is fixed and the market demand depends on price, promotion efforts and carbon reduction and drainage. Ma *et al.* [16] took the three-echelon cold chain system composed of supplier -TPLSP- retailer as the research object, analyzed the influence of carbon transaction price, consumer's freshness and environmental preference on the decision-making and benefit of the system, and designed a contractual incentive mechanism combining wholesale price and two-part tariff. In addition, some scholars have studied the inventory management of fresh agricultural products under the consideration of fresh-keeping carbon emissions [17–19].

From the existing research, on the one hand, there are abundant research results on the fresh-keeping decision-making and coordination of fresh agricultural products, but there are few studies on the fresh-keeping decision-making and coordination of fresh agricultural products considering carbon emissions. On the other hand, although some studies have noticed the relationship between fresh-keeping and carbon emission, they only take fresh-keeping level as an influencing parameter to study the carbon emission reduction and inventory management of enterprises. To sum up, it seems that the impact of carbon cap-and-trade policy on the fresh-keeping efforts of suppliers of fresh agricultural products is not clear in the current research, and there is also a lack of research on how to motivate fresh agricultural products suppliers to improve their fresh-keeping efforts under the carbon cap-and-trade policy, so as to achieve the triple goals of satisfying consumers' consumption preferences, controlling carbon emissions and achieving optimal supply chain benefits at the same time. Therefore, to fulfill the gaps above, this paper will take the two-echelon supply chain of fresh agricultural products led by suppliers as the research object. On the one hand, it will study the influence of carbon cap-and-trade policy on suppliers' fresh-keeping decisions. On the other hand, it will design contracts under the carbon cap-and-trade policy to encourage suppliers to increase their fresh-keeping efforts and realize the coordination of the supply chain.

The structure of the rest of this paper is as follows: Section 2 explains the basic assumptions of the model construction and the symbols used in this paper; Section 3 constructs game models in different situations and compares the optimal results of different models; Section 4 is numerical analysis; Section 5 summarizes the conclusions and management enlightenment of this study, and points out the shortcomings of the study.

## 2 Assumption and symbol description

This paper considers a two-echelon supply chain consisting of a fresh agricultural product supplier *g* and a fresh agricultural product retailer *s*, and the supplier is dominant. Retailers buy fresh agricultural products from suppliers, and suppliers are responsible for stocking, transportation and distribution services. Due to the perishable and perishable nature of fresh agricultural products, suppliers need to use refrigeration equipment to keep freshness during storage and transportation. In other words, the supplier is responsible for the preservation of fresh agricultural products.

The main assumptions and symbols of this paper are described as follows.

1. The supplier's unit cost of producing fresh agricultural products is $c$, and the fresh agricultural products are sold to retailers at the unit wholesale price $w$, and the retailers sell them to consumers at the unit price $p$;

2. The market demand is influenced by both price and consumer's preference for freshness, which is expressed as $D = d-bp+k\alpha$. $d$ represents the potential market demand, $k$ represents the sensitivity coefficient of consumers to the freshness of fresh agricultural products ($k>0$), and $b$ represents the consumer's sensitivity coefficient to price ($b>0$);

3. The cost related to fresh-keeping is expressed as $\frac{1}{2}\mu\alpha^2$, $\mu(\mu > 0)$ is the fresh-keeping cost coefficient and $\alpha(\alpha>0)$ is the fresh-keeping effort level of suppliers;

4. $E$ is the carbon emission limit given by the government to suppliers, and $\beta$ is the carbon trading price;

5. Referring to the description of fresh-keeping and carbon emissions by Bai *et al.* [15] and Ma *et al.* [16], the carbon emission generated by fresh-keeping efforts is $\frac{1}{2}\gamma\alpha^2$, $\gamma$ is the carbon emission coefficient. And for the purpose of simply studying the relationship between fresh-keeping and carbon emissions, the carbon emissions generated in other processes are not considered;

6. Suppliers make low-carbon efforts by introducing carbon emission reduction technology, reasonably setting the temperature of cold storage and training personnel, which can reduce the carbon emission coefficient by $e(e<\gamma)$. At this time, the reduced carbon emission is expressed as $\frac{1}{2}e\alpha^2$. And carbon emission reduction cost paid by the supplier is $\frac{1}{2}ne\alpha^2$, $n$ represents the carbon emission reduction cost coefficient, $e$ is the carbon emission reduction coefficient.

## 3 Model construction

### 3.1 Basic model

**3.1.1 Decision mode without carbon cap-and-trade.** When the government does not implement the carbon cap-and-trade policy, the revenue functions of the fresh agricultural products supplier and the fresh agricultural products retailer are:

$$\pi_g = (w - c)(d - bp + k\alpha) - \frac{1}{2}\mu\alpha^2 \tag{1}$$

$$\pi_s = (p - w)(d - bp + k\alpha)$$

First, make the first-order derivative of the retailer's revenue with respect to $p$ equal to zero, we can obtain $p = \frac{d+bw+k\alpha}{2b}$. Bring it into Formula (1), and make the first-order derivatives of $\alpha$ and $w$ equal to zero respectively, we can obtain $w^{*1} = \frac{2\mu(d+bc)-k^2c}{4\mu b-k^2}$, $\alpha^{*1} = \frac{k(d-bc)}{4\mu b-k^2}$, $p^{*1} = \frac{\mu(3d+bc)-k^2c}{4\mu b-k^2}$. The Hessian matrix $H_1(\alpha, w)$ of Formula (1) is:

$$H_1(\alpha, w) = \begin{bmatrix} -\mu & k/2 \\ k/2 & -b \end{bmatrix}$$

The first-order principal sub-formula of the matrix is $D_1 = -\mu<0$; second-order principal sub-formula is $D_2 = \mu b - \frac{k^2}{4}$. Because $\alpha^2>0$ and $d-bc>0$, obviously, $4\mu b-k^2>0$, that is $D_2>0$.

Therefore, $w^{*1}$ and $\alpha^{*1}$ can be obtained at the same time. At this time, the retailer's income is $\pi_s^{*1} = \frac{b\mu^2(d-bc)^2}{(4\mu b - k^2)^2}$, the supplier's income is $\pi_g^{*1} = \frac{\mu(d-bc)^2}{2(4\mu b - k^2)}$, the total revenue of supply chain is $\pi_Z^{*1} = \frac{\mu(d-bc)^2(6\mu b - k^2)}{2(4\mu b - k^2)^2}$.

### 3.1.2 Decision mode with carbon cap-and-trade.

When the government implements the carbon cap-and-trade policy, the revenue functions of suppliers and retailers are:

$$\pi_g = (w - c)(d - bp + k\alpha) - \frac{1}{2}\mu\alpha^2 - \frac{1}{2}ne\alpha^2 - \beta(\frac{1}{2}\gamma\alpha^2 - \frac{1}{2}e\alpha^2 - E) \tag{2}$$

$$\pi_s = (p - w)(d - bp + k\alpha)$$

Make the first-order derivative of the retailer's revenue with respect to $p$ equal to zero, we can obtain $p = \frac{d + bw + k\alpha}{2b}$. Bring it into Formula (2), and make the first-order derivatives of $\alpha$ and $w$ equal to zero respectively, then we can obtain $w^{*2} = \frac{2A(d+bc) - k^2c}{4Ab - k^2}, \alpha^{*2} = \frac{k(d-bc)}{4Ab - k^2}$, $p^{*2} = \frac{A(3d+bc) - k^2c}{4Ab - k^2}$ $(A = \mu + ne + \beta\gamma - \beta e)$. The Hessian matrix $H_2(\alpha, w)$ of Formula (2) is:

$$H_2(\alpha, w) = \begin{bmatrix} -A & k/2 \\ k/2 & -b \end{bmatrix}$$

The first-order principal sub-formula of the matrix is $D_1 = -A < 0$; second-order principal sub-formula is $D_2 = Ab - \frac{k^2}{4}$. Because $\alpha^2 > 0$ and $d - bc > 0$, obviously, $4Ab - k^2 > 0$, that is $D_2 > 0$. Therefore, $w^{*2}$ and $\alpha^{*2}$ can be obtained at the same time. At this time, the retailer's revenue is $\pi_s^{*2} = \frac{bA^2(d-bc)^2}{(4Ab - k^2)^2}$, the supplier's revenue is $\pi_g^{*2} = \frac{A(d-bc)^2}{2(4Ab - k^2)} + \beta E$, the total revenue of supply chain is $\pi_Z^{*2} = \frac{A(d-bc)^2(6Ab - k^2)}{2(4Ab - k^2)^2} + \beta E$.

### 3.1.3 Comparison of the optimal fresh-keeping decision and revenue under two modes.

*Proposition 1*: The fresh-keeping efforts of suppliers and the profits of retailers with carbon cap-and-trade are lower than those without carbon cap-and-trade.

*Proof*: By making a difference in the optimal fresh-keeping decision of suppliers with and without carbon cap-and-trade policy, we can get:

$$\alpha^{*2} - \alpha^{*1} = \frac{k(d - bc)}{4Ab - k^2} - \frac{k(d - bc)}{4\mu b - k^2}$$

Due to $A = \mu + ne + \beta\gamma - \beta e$ and $y > e$, so $A > \mu$. That is, $4Ab - k^2 > 4\mu b - k^2$. Then $\alpha^{*2} - \alpha^{*1} < 0$.

Make a subtraction between the income of fresh agricultural products retailers in two modes, we can get:

$$\begin{aligned}
\pi_s^{*2} - \pi_s^{*1} &= \frac{bA^2(d - bc)^2}{(4Ab - k^2)^2} - \frac{b\mu^2(d - bc)^2}{(4\mu b - k^2)^2} \\
&= b(d - bc)^2 \left( \frac{A}{4Ab - k^2} + \frac{\mu}{4\mu b - k^2} \right) \left( \frac{\mu k^2 - Ak^2}{(4Ab - k^2)(4\mu b - k^2)} \right) < 0
\end{aligned}$$

*Proposition 2*: When $\frac{(d-bc)^2}{2} \left( \frac{\mu k^2 - Ak^2}{(4Ab - k^2)(4\mu b - k^2)} \right) + \beta E > 0$, the fresh agricultural products supplier's income under carbon cap-and-trade policy is higher than that without cap-and-trade policy; when $\frac{(d-bc)^2}{2} \left( \frac{\mu k^2 - Ak^2}{(4Ab - k^2)(4\mu b - k^2)} \right) + \beta E < 0$, the fresh agricultural products supplier's income under carbon cap-and-trade policy is lower than that without cap-and-trade policy.

*Proof*: By making a subtraction between the income of fresh agricultural products suppliers in two cases, we can get:

$$\pi_g^{*2} - \pi_g^{*1} = \frac{A(d-bc)^2}{2(4Ab-k^2)} - \frac{\mu(d-bc)^2}{2(4\mu b-k^2)} + \beta E = \frac{(d-bc)^2}{2}\left(\frac{\mu k^2 - Ak^2}{(4Ab-k^2)(4\mu b-k^2)}\right) + \beta E$$

Obviously, when $\frac{(d-bc)^2}{2}\left(\frac{\mu k^2 - Ak^2}{(4Ab-k^2)(4\mu b-k^2)}\right) + \beta E > 0$, there is $\pi_g^{*2} - \pi_g^{*1} > 0$; when $\frac{(d-bc)^2}{2}\left(\frac{\mu k^2 - Ak^2}{(4Ab-k^2)(4\mu b-k^2)}\right) + \beta E < 0$, there is $\pi_g^{*2} - \pi_g^{*1} < 0$.

## 3.2 Contract coordination mode under carbon cap-and-trade

The implementation of carbon cap-and-trade policy will reduce the fresh-keeping efforts of suppliers, and may bring about the reduction of supply chain revenue. Then, under the implementation of carbon cap-and-trade, how should we design contracts to coordinate suppliers' fresh-keeping decisions and ensure the benefits of suppliers and retailers of fresh agricultural products? This is the next problem to be solved.

**3.2.1 Cost-sharing contract.** Under the cost sharing contract, retailers are willing to bear the fresh-keeping cost and carbon emission cost with a proportion of $\emptyset(0<\emptyset<1)$. At this time, the revenue functions of suppliers and retailers are:

$$\pi_g = (w-c)(d - bp + k\alpha) - (1-\emptyset)(\frac{1}{2}\mu\alpha^2 + \frac{1}{2}ne\alpha^2 + \beta(\frac{1}{2}\gamma\alpha^2 - \frac{1}{2}e\alpha^2 - E)) \tag{3}$$

$$\pi_s = (p-w)(d - bp + k\alpha) - \emptyset(\frac{1}{2}\mu\alpha^2 + \frac{1}{2}ne\alpha^2 + \beta(\frac{1}{2}\gamma\alpha^2 - \frac{1}{2}e\alpha^2 - E)) \tag{4}$$

Using the same solution method as that in Section 3.1, optimal decisions under the cost-sharing contract mode can be obtained:

$w^{*3} = \frac{2A(1-\emptyset)(d+bc)-k^2c}{4Ab(1-\emptyset)-k^2}, \alpha^{*3} = \frac{k(d-bc)}{4Ab(1-\emptyset)-k^2}, p^{*3} = \frac{A(1-\emptyset)(3d+bc)-k^2c}{4Ab(1-\emptyset)-k^2}$. Bringing the results into Formula (3) and Formula (4), we can get the income function expressions about the sharing ratio. The income of fresh agricultural products suppliers is: $\pi_g^{3} = \frac{A(1-\emptyset)(d-bc)^2}{2(4Ab(1-\emptyset)-k^2)} + (1-\emptyset)\beta E$. The income of fresh agricultural products retailers is:

$$\pi_s^{3} = \frac{A(d-bc)^2(2Ab(1-\emptyset)^2 - \emptyset k^2)}{2(4Ab(1-\emptyset) - k^2)^2} + \emptyset\beta E. \tag{5}$$

Find the second-order derivative of Formula (5) with respect to $\emptyset : \frac{\partial^2 \pi_s^{3}}{\partial \emptyset^2} = \frac{A(d-bc)^2 8Abk^2(k^2-4Ab(1-\emptyset))}{2(4Ab(1-\emptyset)-k^2)^4}$. Since $4Ab(1-\emptyset)-k^2>0$, then, $\frac{\partial^2 \pi_s^{3}}{\partial \emptyset^2} < 0$. So, the optimal cost-sharing ratio $\emptyset^* = \frac{k^2}{8Ab}$ can be obtained.

At this time, the income of the supplier is: $\pi_g^{*3} = \frac{A(d-bc)^2(8Ab-k^2)}{8Ab(8Ab-3k^2)} + \frac{8Ab-k^2}{8Ab}\beta E$. The income of the retailer is: $\pi_s^{*3} = \frac{A(d-bc)^2(8Ab+k^2)}{16Ab(8Ab-3k^2)} + \frac{k^2}{8Ab}\beta E$.

*Proposition 3* Under the carbon cap-and-trade policy, compared with the no-contract mode, the fresh-keeping efforts of suppliers of fresh agricultural products are increased under the cost-sharing contract.

*Proof*: The difference between the fresh-keeping efforts under the cost-sharing contract mode and the non-contract mode is:

$$\alpha^{*3} - \alpha^{*2} = \frac{4Ab\emptyset k(d - bc)}{(4Ab(1 - \emptyset) - k^2)(4Ab - k^2)} > 0$$

*Proposition 4* The coordination effect of cost-sharing contract on supply chain revenue is related to $\beta$ and $E$. When certain conditions are met, it can realize Pareto improvement on the income of fresh agricultural products retailers and suppliers.

*Proof*: The difference between the revenue of suppliers under the cost-sharing contract mode and the non-contract mode is:

$$\pi_g^{*3} - \pi_g^{*2} = \frac{k^4 A(d - bc)^2}{8Ab(8Ab - 3k^2)(4Ab - k^2)} - \frac{k^2}{8Ab}\beta E$$

To make $\pi_g^{*3} - \pi_g^{*2} > 0$, the condition $\beta E < \frac{k^2 A(d - bc)^2}{(8Ab - 3k^2)(4Ab - k^2)}$ need to be met.

The difference between the revenue of retailers under the cost-sharing contract mode and the non-contract mode is:

$$\pi_s^{*3} - \pi_s^{*2} = \frac{A(d - bc)^2}{16Ab(8Ab - 3k^2)(4Ab - k^2)^2} + \frac{k^2}{8Ab}\beta E > 0$$

In general, when $\beta E < \frac{k^2 A(d - bc)^2}{(8Ab - 3k^2)(4Ab - k^2)}$, the cost-sharing contract can realize Pareto improvement of retailer's and supplier's revenue.

**3.2.2 Two-part pricing contract.** Under the two-part pricing contract, the supplier offers a lower wholesale price $w$ to the retailer and charges a fixed remuneration $F$. The order of decision-making is: the supplier first decides the fresh-keeping effort level $\alpha$ based on the principle of maximizing benefits, and provides the fresh agricultural product retailer with a two-part pricing contract ($w, F$). The retailer chooses to accept or reject it. If the retailer accepts, the contract is established, and the retailer places an order and determines the retail price. At this time, revenue functions of the supplier and the retailer are:

$$\pi_g = (w - c)(d - bp + k\alpha) - \frac{1}{2}\mu\alpha^2 - \frac{1}{2}ne\alpha^2 - \beta(\frac{1}{2}\gamma\alpha^2 - \frac{1}{2}e\alpha^2 - E) + F \tag{6}$$

$$\pi_s = (p - w)(d - bp + k\alpha) - F \tag{7}$$

Find the first-order derivative of Formula (6) with respect to $\alpha$ and make it equal to zero, we can obtain $\alpha = \frac{(w-c)k}{A}$. Bring it into Formula (7), find the first-order derivative of Formula (7) with respect to $p$ and make it equal to zero, we can get $p = \frac{A(d+bw)+k(w-c)}{2bA}$. At this point, the problem turns into:

$$max\pi_s(w, F) = \frac{(d - bw)(w - c)}{2} + F + \beta E \tag{8}$$

$$s.t.\pi_g = \frac{(A(d - bw) + k^2(w - c))^2}{4A^2 b} - F \geq \frac{bA^2(d - bc)^2}{(4Ab - k^2)^2}$$

Solve the constraints, bring it into Formula (8) and find its second-order derivative with respect to $w$ : $\frac{\partial^2 \pi_g}{\partial w^2} = \frac{k^2(k^2 - 2Ab) - A^2 b^2}{2A^2 b} < 0$. Then, make the first-order derivative of Formula (8)

with respect to $w$ equal to zero, we can get the optimal solution of wholesale price:
$w^{*4} = \frac{k^2A(d+bc)+(A^2b^2-k^4)c}{A^2b^2+k^2(2Ab-k^2)}$.

At this point, $F = \frac{A^2b(Ab+k^2)^2(d-bc)^2}{4(A^2b^2+k^2(2Ab-k^2))^2} - \frac{bA^2(d-bc)^2}{(4Ab-k^2)^2}$, $\alpha^{*4} = \frac{(d-bc)k^3}{A^2b^2-k^4+2Abk^2}$, $p^{*4} = \frac{A(3d+bc)k^2-2ck^4+A^2b(d+bc)}{2(A^2b^2-k^4+2Abk^2)}$,

$\pi_s^{*4} = \frac{bA^2(d-bc)^2}{(4Ab-k^2)^2}$, $\pi_g^{*4} = \frac{(Ab+2k^2)A(d-bc)^2}{4(A^2b^2-k^4+2Abk^2)} - \frac{bA^2(d-bc)^2}{(4Ab-k^2)^2} + \beta E$.

*Proposition 5* Under certain conditions, the two-part pricing contract can realize Pareto improvement of retailers' and suppliers' income, and can improve the fresh-keeping efforts of suppliers.

*Proof*: Firstly, it is obvious that the income of fresh agricultural products retailers under the two-part pricing contract is greater than or equal to the income under the non-contract mode. Secondly, the difference of the supplier's income under the two-part pricing contract mode and non-contract mode is:

$$\pi_g^{*4} - \pi_g^{*2} = \frac{A(d-bc)^2(4A^3b^3 + Abk^4 + 2A^2b^2k^2)}{4(A^2b^2 - k^4 + 2Abk^2)(4Ab - k^2)^2} > 0$$

The difference of fresh-keeping efforts between two-part pricing contract mode and non-contract mode is:

$$\alpha^{*4} - \alpha^{*2} = \frac{Abk(d-bc)(2k^2 - Ab)}{(A^2b^2 + k^2(2Ab - k^2))(4Ab - k^2)}$$

To make $\alpha^{*4} > \alpha^{*2}$, there must be $2k^2 - Ab > 0$, that is, $k^2 > \frac{Ab}{2}$. And because $2Ab - k^2 > 0$, $k^2$ must meet the condition $k^2 \in \left(\frac{Ab}{2}, 2Ab\right)$.

To sum up, when $\frac{Ab}{2} < k^2 < 2Ab$, the two-part pricing contract can realize the Pareto improvement of supply chain income, and can encourage suppliers to increase their fresh-keeping efforts.

**3.2.3 Comparison between cost-sharing contract and two-part pricing contract.** Under certain conditions, that is, satisfying $\beta E < \frac{k^2A(d-bc)^2}{(8Ab-3k^2)(4Ab-k^2)}$ and $\frac{Ab}{2} < k^2 < 2Ab$ at the same time, both the cost-sharing contract and the two-part pricing contract can realize the coordination of the fresh agricultural products supply chain. Further comparing the two contracts, we can find the following conclusions.

*Proposition 6* When the condition $4Abk^2 - k^4 - 2A^2b^2 > 0$ is further satisfied, the coordination effect of the two-part pricing contract on the supplier's fresh-keeping efforts is better than that of the cost-sharing contract.

*Proof*: The difference between the fresh-keeping efforts of suppliers under two contracts is:

$$\alpha^{*4} - \alpha^{*3} = \frac{(4Abk^2 - k^4 - 2A^2b^2)k(d-bc)}{(A^2b^2 + k^2(2Ab - k^2))(8Ab - 3k^2)}$$

When $4Abk^2 - k^4 - 2A^2b^2 > 0$, $\alpha^{*4} - \alpha^{*3} > 0$. That is, the fresh-keeping efforts of suppliers under the two-part pricing contract is higher than cost-sharing contract.

*Proposition 7* When both contracts are valid, the retailer's revenue under the cost-sharing contract is greater than that of the two-part pricing contract, and the total revenue of supply chain and suppliers under the two-part pricing contract are greater than that of the cost-sharing contract.

*Proof*: The difference between the total supply chain revenue under two contracts is:

$$\pi_Z{}^{*4} - \pi_Z{}^{*3} = \frac{A(d-bc)^2(8A^3b^3 + 5A^2b^2k^2 + k^4(2Ab - k^2)}{16Ab(8Ab - 3k^2)(A^2b^2 + k^2(2Ab - k^2))} > 0$$

The difference between the retailer's revenue under the two contracts is:

$$\pi_s{}^{*4} - \pi_s{}^{*3} = -\frac{A(d-bc)^2}{16Ab(8Ab - 3k^2)(4Ab - k^2)^2} - \frac{k^2}{8Ab}\beta E < 0$$

Since $\pi_Z{}^{*4} - \pi_Z{}^{*3} > 0$ and $\pi_s{}^{*4} - \pi_s{}^{*3} < 0$, there must be $\pi_g{}^{*4} - \pi_g{}^{*3} > 0$.

*Inference 3* For suppliers, the revenue under the two-part pricing contract is always higher than that under the cost-sharing contract. However, under certain conditions ($k^2 < \frac{Ab}{2}$ or $k^2 > 2Ab$), the incentive of the two-part pricing contract to fresh-keeping efforts fails. At this time, the coordination effect of the cost-sharing contract is better.

## 4 Numerical analysis

In order to further understand the influence of the parameter changes on the supplier's fresh-keeping decision and the profit of the supply chain, and compare the coordination effect of the two contracts more clearly, a numerical analysis is carried out below. According to the constraints calculated above, the parameters are assigned as: $d = 50$, $b = 1$, $c = 10$, $k = 2$, $\mu = 4$, $n = 2$, $\gamma = 2$, $e = 0.5$, $\beta = 1$, $E = 40$. The optimal decisions and benefits under different modes are shown in Table 1.

It can be clearly seen from the results that: First of all, the fresh-keeping efforts, retailer's revenue and total revenue of the fresh agricultural supply chain under the carbon cap-and-trade policy are lower than those without the carbon cap-and-trade policy. Secondly, when the carbon cap-and-trade policy is implemented, under the coordination of the cost-sharing contract and the two-part pricing contract, the fresh-keeping efforts of suppliers have been improved, and the total income of supply chain has also been improved. Both the cost-sharing contract and the two-part pricing contract can realize the coordination of the fresh agricultural supply chain. Further comparing the two contracts, it can be found that the fresh-keeping efforts and benefits of suppliers under the two-part pricing contract are higher than those under the cost-sharing contract. While for retailers, the benefits under the cost-sharing contract are higher than those under the two-part pricing contract. Next, we will discuss the impact of different parameter changes on fresh-keeping decisions and revenues.

**Table 1. Optimal decision-making and revenue under different modes.**

| Variables | No carbon cap-and -trade policy | Carbon cap-and-trade policy | | |
|---|---|---|---|---|
| | | No contract | Cost sharing contract | Two-part pricing contract |
| $\alpha$ | 6.67 | 3.64 | 4 | 4.09 |
| $p$ | 50 | 45.45 | 46 | 40.73 |
| $w$ | 36.67 | 33.64 | 34 | 23.29 |
| $\pi_g$ | 266.67 | 276.36 | 276.92 | 471.79 |
| $\pi_s$ | 177.78 | 139.67 | 143.08 | 139.67 |
| $\pi_Z$ | 444.45 | 416.03 | 420 | 611.46 |

### 4.1 Influence of *k* on fresh-keeping decisions and revenues

Fig 1 shows the influence of consumers' sensitivity coefficient to freshness on fresh-keeping decisions and revenues. With the increase of consumers' sensitivity coefficient to the freshness of fresh agricultural products, the fresh-keeping efforts of suppliers, the retail price of fresh agricultural products, the income of suppliers and the income of retailers are all increasing. When consumers' preference for the freshness increases, suppliers will increase their fresh-keeping efforts to meet consumers' demand. At this time, the price of fresh agricultural products will increase, and accordingly, the profits of suppliers and retailers will both increase.

Comparing the fresh-keeping efforts and benefits under the two contracts, we find that the smaller the consumers' preference for freshness, the better the coordination effect of cost-sharing contract; the greater the consumers' preference for freshness, the better the coordination effect of two-part pricing contract. Specifically, when consumers are less sensitive to the freshness of fresh agricultural products, the cost-sharing contract can encourage suppliers to increase their fresh-keeping efforts and increase the profits of suppliers and retailers. At this time, although the two-part pricing contract can achieve Pareto improvement of supply chain profits, it cannot motivate suppliers to improve their fresh-keeping efforts. With the increase of consumers' preference for freshness of fresh agricultural products, the incentive effect of two-part pricing contract on suppliers' fresh-keeping efforts is gradually emerging. This is because retailers directly share some suppliers' fresh-keeping costs, which can directly motivate suppliers to increase their fresh-keeping efforts; however, the essence of the two pricing contracts is the secondary distribution of sales revenue, and the coordination effect depends on the consumer market reaction, so it is more sensitive to the changes of consumers' freshness preference.

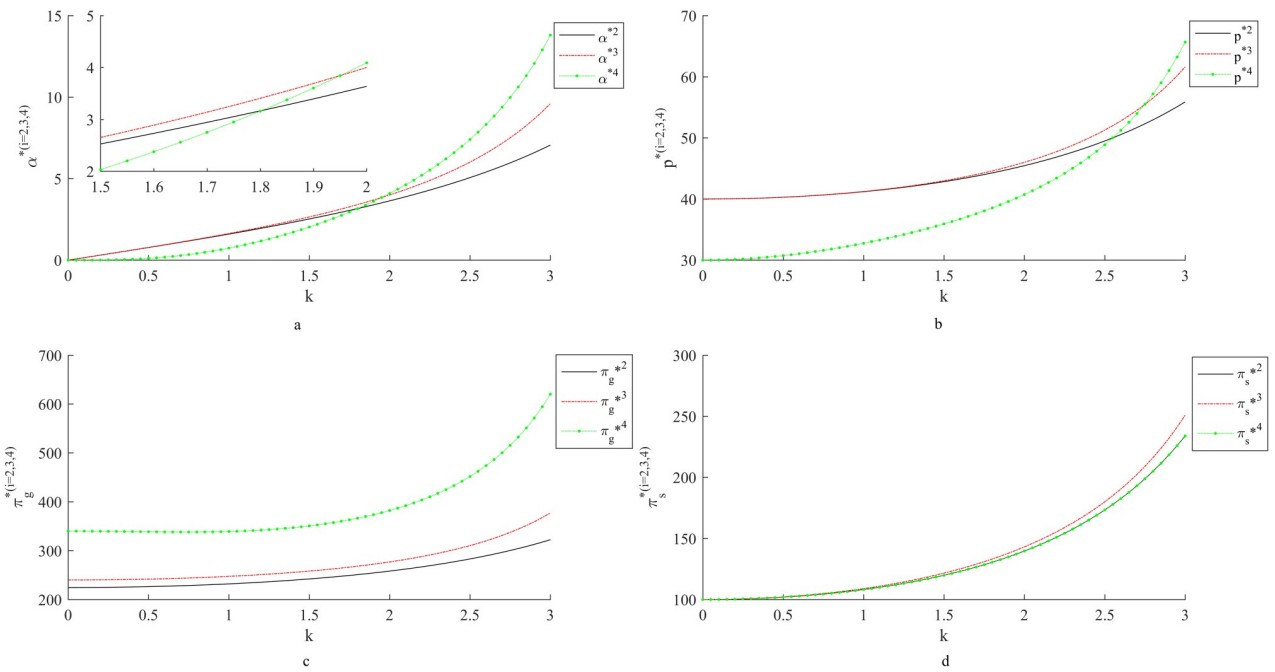

**Fig 1. Influence of *k* on fresh-keeping decisions and revenues.**

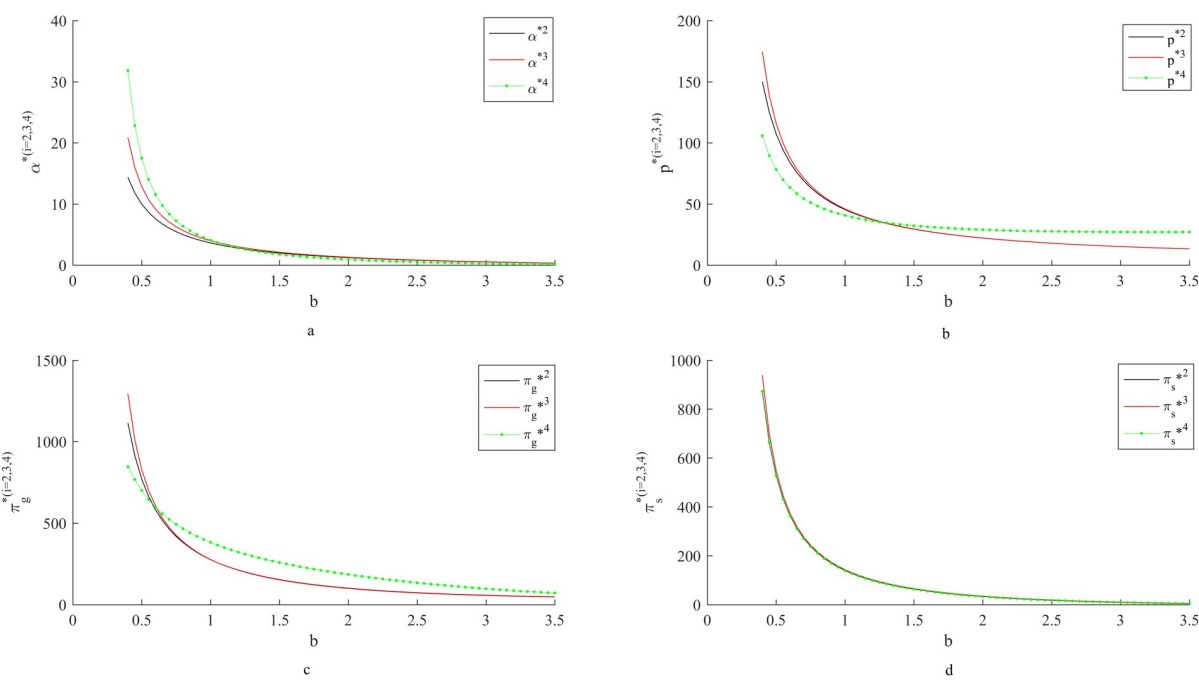

**Fig 2. Influence of *b* on fresh-keeping decisions and revenues.**

## 4.2 Influence of *b* on fresh-keeping decisions and revenues

As can be seen from Fig 2: With the increase of the sensitivity coefficient of consumers to the price of fresh agricultural products, the fresh-keeping efforts of fresh agricultural products suppliers, the retail price of fresh agricultural products, the income of suppliers and the income of retailers all decrease. When consumers are sensitive to the price of fresh agricultural products, suppliers will reduce the cost of fresh-keeping by reducing their fresh-keeping efforts, thus reducing the retail price. At this time, the marginal profit of fresh agricultural products is low, resulting in a decrease in income.

When consumers are less sensitive to the price of fresh agricultural products, both the cost-sharing contract and the two-part pricing contract can encourage suppliers to increase their efforts to keep fresh and increase supply chain revenue. But the fresh-keeping efforts and supply chain benefits under the two-part pricing contract are higher than those under the cost-sharing contract. However, with the increase of consumers' sensitivity to price, the incentive effect of the two-part pricing contract on fresh-keeping efforts gradually fails. This is because when consumers do not pay attention to the price, suppliers will increase their fresh-keeping efforts to meet consumers' preference for freshness. At the same time, because the retailer does not need to consider the preservation cost when pricing under the two-part pricing contract, the retail price under the two-part pricing contract is lower than that under the cost-sharing contract. That is to say, the supply chain under the coordination of two-part pricing contract can provide consumers with fresh agricultural products with high freshness and favorable prices, so the benefits of the supply chain under the coordination of two-part pricing contract are higher than those under the cost-sharing contract. When consumers pay more attention to the price of fresh agricultural products, they are unwilling to pay higher prices for fresh agricultural products with high freshness. At this time, the suppliers under the two-part pricing contract will continue to reduce their fresh-keeping efforts and the contract will be invalid. However, under the cost-sharing contract, retailers share part of the fresh-keeping cost, which

can reduce the supplier's fresh-keeping cost, thus encouraging suppliers to increase their fresh-keeping efforts.

### 4.3 Influence of $\mu$, $\gamma$ on fresh-keeping decisions and revenues

As shown in Fig 3: Firstly, with the increase of fresh-keeping cost coefficient and carbon emission coefficient, the fresh-keeping efforts, the retail price, the supplier's income and the retailer's income are all decreasing. The larger fresh-keeping cost coefficient and carbon emission coefficient mean that the fresh-keeping efforts will bring more fresh-keeping costs and carbon transaction costs. Therefore, suppliers will choose to reduce fresh-keeping efforts to reduce costs, which will lead to a decline in retail prices and revenue.

Secondly, in a certain range, both the two-part pricing contract and the cost-sharing contract can coordinate the supply chain, but there are differences. When the fresh-keeping cost coefficient and carbon emission coefficient are small, due to the different pricing mechanisms, compared with the cost-sharing contract, the supply chain under the two-part pricing contract can occupy more markets at lower price with the same fresh-keeping efforts, thus obtaining more benefits. When the fresh-keeping cost coefficient and carbon emission coefficient are large, similarly, because the retailer does not consider the fresh-keeping cost when pricing under the two-part pricing contract, that is to say, the marginal cost of the supplier's fresh-keeping efforts under the two-part pricing contract is greater than the marginal income, so the incentive effect of the contract on fresh-keeping efforts is invalid. However, the cost-sharing contract can reduce the supplier's fresh-keeping cost, thus encouraging the supplier to improve the fresh-keeping efforts, thereby increasing the income. At the same time, because the retailer can reasonably transfer the fresh-keeping cost to the retail price when pricing, the retailer's income will also increase.

### 4.4 Influence of $n$, $e$ on fresh-keeping decisions and revenues

As can be seen from Fig 4, when the cost coefficient of carbon emission reduction is small, with the increase of carbon emission reduction coefficient, suppliers' fresh-keeping efforts will

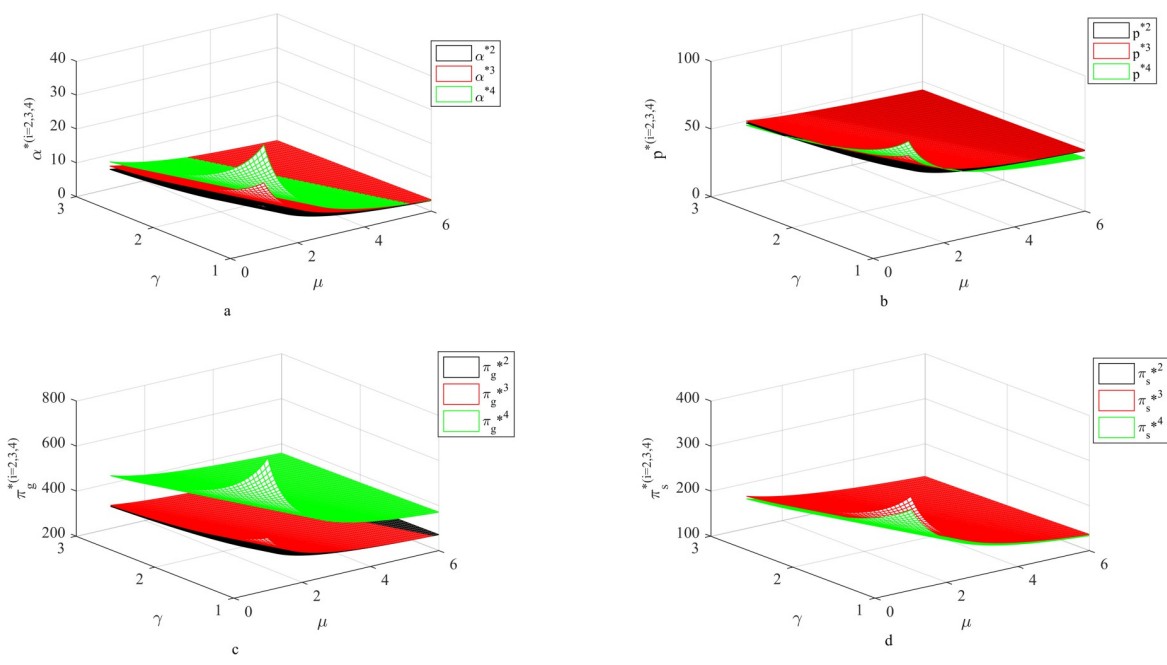

**Fig 3. Influence of $\mu$, $\gamma$ on fresh-keeping decisions and revenues.**

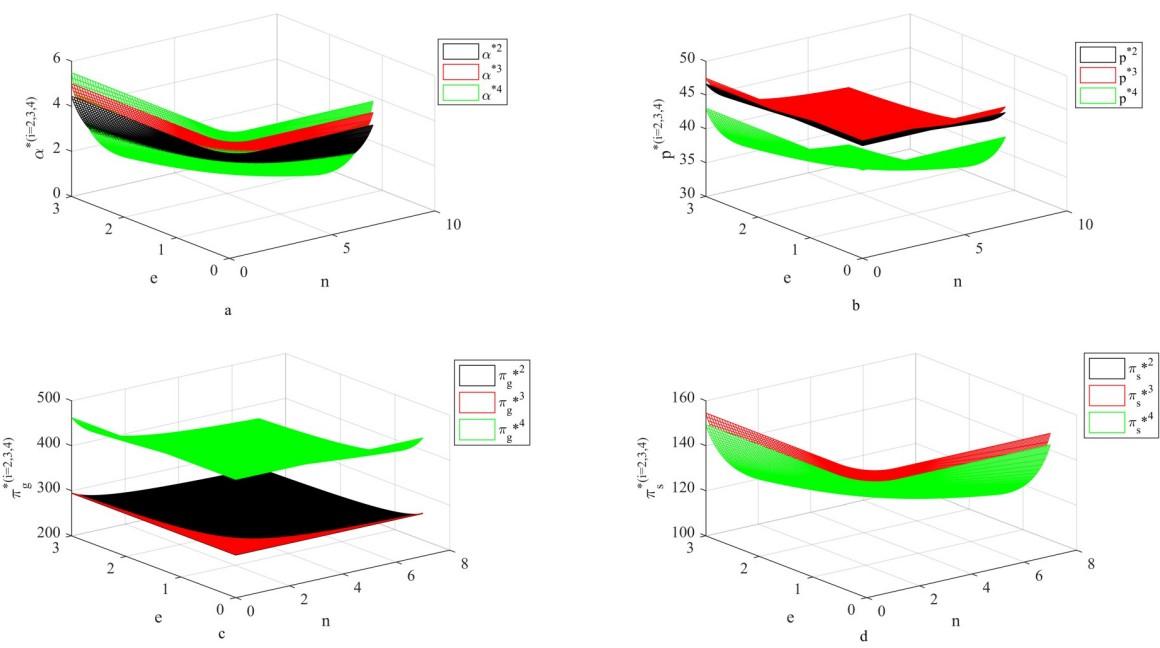

**Fig 4. Influence of $n$, $e$ on fresh-keeping decisions and revenues.**

increase. At this time, suppliers improve their fresh-keeping efforts, which can not only meet consumers' preference for freshness and bring more market benefits, but also pay less carbon emission reduction costs to achieve better carbon emission reduction effects and reduce carbon transaction costs. In addition, we can also see that when the cost of carbon reduction is low, the two-part pricing contract is more effective for the coordination of supply chain. However, when the cost of carbon emission reduction is high, the two-part pricing contract loses its effectiveness. And at this time, directly sharing the cost can encourage suppliers to improve their fresh-keeping efforts.

## 4.5 Influence of $\beta$, $E$ on fresh-keeping decisions and revenues

Fig 5 shows that the supplier's fresh-keeping effort decision has nothing to do with carbon quota, but only with carbon transaction price. When the carbon quota is fixed, the higher the carbon transaction price, the lower the supplier's fresh-keeping efforts, the lower the retail price and the smaller the income. Because with the increase of carbon transaction price, increasing fresh-keeping efforts will bring more carbon transaction costs to suppliers, so suppliers choose to reduce fresh-keeping efforts, and at this time, the retail price decreases and the income decreases. However, when the carbon transaction price is high, suppliers choose to greatly reduce the preservation efforts, which brings redundant carbon emission quotas. At this time, the supplier can gain profits by selling the carbon quota, so although the supplier's fresh-keeping efforts are reduced, the supplier's income is still increased.

We can also see that under a certain carbon limit, when the carbon transaction price is small, although both contracts are effective, the coordination effect of the two-part pricing contract is better; when the carbon transaction price is high, the coordination effect of two-part pricing contract on fresh-keeping efforts fails, and the coordination effect of cost-sharing contract is better. When the carbon trading price is small, fresh agricultural products under the coordination of two-part pricing contract have higher freshness and lower price, so the two-part pricing contract is better; however, when the carbon transaction price is high, the

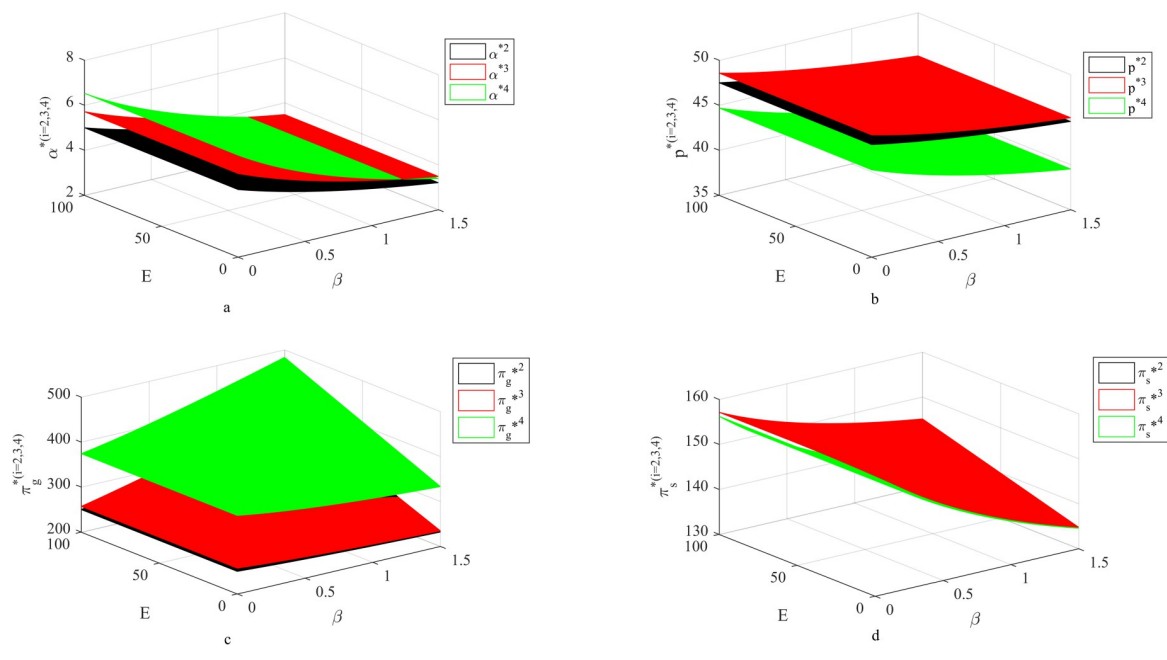

**Fig 5. Influence of $\beta$, $E$ on fresh-keeping decisions and revenues.**

carbon transaction cost for suppliers to increase their fresh-keeping efforts is high, and it is more effective to share the cost directly at this time.

## 5 Conclusion

This paper mainly studies the influence of carbon cap-and-trade policy on the fresh-keeping decision of suppliers of fresh agricultural products, and how to design contracts to coordinate the supply chain of fresh agricultural products under carbon cap-and-trade policy. First of all, we built a Stackelberg game model of fresh agricultural products supply chain on fresh-keeping decision without carbon cap-and-trade policy, which provided a comparative benchmark for the subsequent model construction. Secondly, we built a Stackelberg game model of fresh agricultural products supply chain on fresh-keeping decision under carbon cap-and-trade policy, and compared it with the previous model. Finally, we designed a cost-sharing contract and a two-part pricing contract for coordination based on the Stackelberg game model of fresh agricultural product supply chain regarding fresh-keeping decision under the carbon cap-and-trade policy, and made a comparative analysis of the scope of action and coordination effect of the two contracts. The main conclusions are as follows.

1. Whether the carbon cap-and-trade policy is implemented or not, suppliers' fresh-keeping efforts are related to consumers' preference for freshness and consumers' sensitivity coefficient to price. When consumers have higher preference for freshness of fresh agricultural products and lower sensitivity to price, that is, when consumers pay more attention to freshness of fresh agricultural products and don't care about the price of products, consumers are willing to pay higher prices for the preservation efforts made by suppliers, and the income of suppliers and retailers increases at this time. When consumers' preference for freshness is low and their sensitivity to price is high, that is, consumers want to buy low-priced fresh agricultural products, but they don't pay much attention to the freshness of fresh agricultural products, suppliers will reduce their fresh-keeping efforts and lower the price of products to get more benefits.

2. Compared with the case that the carbon cap-and-trade policy is not implemented, the implementation of the carbon cap-and-trade policy will bring carbon emission costs to suppliers, which is not conducive to suppliers to increase their fresh-keeping efforts. Because the higher the carbon transaction price, the higher the carbon transaction cost brought by the supplier's fresh-keeping efforts. At this point, suppliers will reduce their fresh-keeping efforts to ensure revenue. In addition, we also find that the impact of carbon cap-and-trade policy on suppliers' income depends on the price of carbon trading and the amount of carbon cap. When the carbon transaction price is too high, the supplier will greatly reduce their fresh-keeping efforts, resulting in carbon emissions below the total carbon quota. At this time, the supplier can obtain income by selling carbon quotas, so the income of the supplier is higher than that without implementing the carbon cap-and-trade policy. When the carbon trading price is not high, the supplier needs to pay both the fresh-keeping cost and the carbon trading cost, so the supplier's income is lower than that without implementing the carbon cap-and-trade policy.

3. Compared with the situation without carbon cap-and-trade policy, the implementation of carbon cap-and-trade policy will reduce the fresh-keeping efforts of fresh agricultural products suppliers, so the hitchhiking income of fresh agricultural products retailers will decrease, which will lead to the decrease of retailers' income.

4. Under the carbon cap-and-trade policy, the smaller the cost coefficient of carbon emission reduction is, the greater the emission reduction coefficient is, that is, when fresh agricultural products suppliers can achieve better carbon emission reduction effect at less cost, their fresh-keeping efforts will be higher.

5. Under the carbon cap-and-trade policy, both the cost-sharing contract and the two-part pricing contract can coordinate the supply chain, but the scope of application and coordination effect are different. First of all, for suppliers, regardless of other parameters, compared with the cost-sharing contract, the supplier's income under the two-part pricing contract is always higher. Secondly, when other parameters remain unchanged and consumers' preference for freshness is too small or too large, compared with the two-part pricing contract, the supplier's fresh-keeping efforts and retailers' income under the cost-sharing contract are higher. Finally, in the parameter range where both contracts are valid, the greater the carbon emission reduction coefficient, the smaller the consumers' price sensitivity, the smaller the fresh-keeping cost coefficient and the smaller the carbon emission coefficient, the supplier's fresh-keeping efforts under the two-part pricing contract are greater than those under the cost-sharing contract.

Our research results have important theoretical and practical significance. We have identified the impact of the cap-and-trade policy on fresh produce suppliers' fresh-keeping efforts. Similar to the conclusions of Ma *et al.* [16], we found that high-carbon transaction price would reduce the fresh-keeping efforts. But our research objects are different. Specifically, Ma *et al.* studied the three- echelon supply chain consisting of suppliers, cold chain logistics enterprises and sales enterprises, but in actual operation, there are still many two-echelon supply chains in which large fresh agricultural products suppliers are responsible for fresh-keeping, such as Bright Dairy Co., Ltd., Hengdu Agricultural Group Co., Ltd., Xinjiang Sanhai Preservation Park Co., Ltd. and so on. The decision-making rules of this type of supply chain are different from those of the three-echelon supply chain. Therefore, our research has enriched the theoretical system of fresh-keeping decision-making research of fresh agricultural products supply chain to a certain extent. In addition, we also designed two contracts, and compared the scope

of application and coordination effect of the two contracts, which can provide more reference for the contract design of the supplier-led fresh agricultural product supply chain.

According to the above conclusions, some management inspirations can be drawn as follows.

1. Under the background of low-carbon development, the government should strengthen the research and development of low-carbon preservation technology and give appropriate subsidies for technology application, which can inhibit the possibility of fresh agricultural products suppliers reducing their fresh-keeping efforts due to the implementation of carbon cap-and-trade policy, improve the freshness of fresh agricultural products and ensure the consumption quality of consumers.

2. When the government formulates the carbon cap-and-trade policy for fresh agricultural products suppliers, it should be noted that too high carbon trading price will inhibit suppliers' fresh-keeping efforts, which is not conducive to meeting consumers' demand for freshness of fresh agricultural products.

3. Suppliers of fresh agricultural products, which occupy a dominant position in the supply chain, should choose the cost-sharing contract and the two-part pricing contract according to different environmental conditions.

Finally, there are still some limitations to be further studied in this paper. On the one hand, the rapid development of fresh e-commerce has greatly changed the structure of the supply chain of fresh agricultural products, and this study has not considered the influence of dual channels on the decision-making of supply chain. On the other hand, this paper does not consider the influence of government subsidies on the decision-making of supply chain.

## Supporting information

**S1 Dataset.**
(DOCX)

## Author Contributions

**Conceptualization:** Guanxin Yao.

**Methodology:** Guanxin Yao.

**Validation:** Yang Yang.

**Writing – original draft:** Yang Yang.

**Writing – review & editing:** Yang Yang, Guanxin Yao.

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
