## [Decision Letter · Decision Letter 0]

20 Feb 2023

PONE-D-22-33342Fresh Keeping Decision and Coordination of Fresh Agricultural Products Supply Chain under Carbon Cap-and-TradePLOS ONE

Dear Dr. Yao,

Thank you for submitting your manuscript to PLOS ONE. After careful consideration, we feel that it has merit but does not fully meet PLOS ONE’s publication criteria as it currently stands. Therefore, we invite you to submit a revised version of the manuscript that addresses the points raised during the review process.

We look forward to receiving your revised manuscript.

Kind regards,

Bo Huang

Academic Editor

PLOS ONE

Journal Requirements:

“The authors gratefully acknowledge financial support from the National Natural Science Foundation of China (Grants No. 72103178), Social Science Foundation of Jiangsu Province (Grant No. 20GLA002), Graduate Research and Innovation Projects of Jiangsu Province (Grant No. SJKY19_2519).”

“GX Y. National Natural Science Foundation of China (Grants No. 72103178);

GX Y. Social Science Foundation of Jiangsu Province (Grant No. 20GLA002);

Y Y. Graduate Research and Innovation Projects of Jiangsu Province (Grant No. SJKY19_2519).”

Reviewers' comments:

Reviewer's Responses to Questions

**Comments to the Author**

1. Is the manuscript technically sound, and do the data support the conclusions?

Reviewer #1: Partly

Reviewer #2: Yes

2. Has the statistical analysis been performed appropriately and rigorously? 

Reviewer #1: Yes

Reviewer #2: Yes

3. Have the authors made all data underlying the findings in their manuscript fully available?

Reviewer #1: No

Reviewer #2: Yes

4. Is the manuscript presented in an intelligible fashion and written in standard English?

Reviewer #1: No

Reviewer #2: Yes

5. Review Comments to the Author

Reviewer #1: Taking the fresh agricultural produce supply chain as the research object, this paper analyzes the impact of carbon cap-and-trade policy on suppliers' preservation decisions and further designs contracts to achieve Pareto improvement of preservation efforts and supply chain benefits.

Under carbon cap-and-trade policy, this paper provides some interesting insights on the suppliers' fresh-keeping decision-making rule and coordination contracts. The discussion on preservation efforts and supply chain benefits under different models is of some practical value. However, I consider that many of the main points in this paper are explained very vaguely. I provide detailed comments and suggestions below.

Major comments：

1. The inspiration and significance of the manuscript should be added after the abstract.

2. In the Introduction, 1st paragraph must support a practical example of which support the problem descriptions. A potential suggestion to the author/s is to support the importance of a title with some practical examples. However, it seems to be so vague that it lacks credibility.

3. In the Introduction, you need to connect the state of the art to your paper goals. Please follow the literature review by a clear and concise state of the art analysis. This should clearly show the knowledge gaps identified and link them to your paper goals. Please reason both the novelty and the relevance of your paper goals. Clearly discuss what the previous studies that you are referring to. What are the Research Gaps/Contributions? Please note that the paper may not be considered further without a clear research gap and novelty of the study.

4. The paper does not provide any background information about the carbon cap-and-trade policy. Hence, it makes the paper not very coherent, and difficult to follow. Please give a specific explanation of the technical term "the carbon cap-and-trade policy" in the manuscript. Additionally, the link between policy implementation and fresh produce preservation should be discussed in depth.

5. In Example Analysis section, all the figures are so vague that it is impossible to distinguish the exact meaning after reading them. Please upload figures in strict accordance with the journal format.

6. The conclusion is very weak. Please make sure your conclusions' section underscores the scientific value-added of your paper, and/or the applicability of your findings/results. Highlight the novelty of your study. In addition to summarizing the actions taken and results, please strengthen the explanation of their significance. It is recommended to use quantitative reasoning comparing with appropriate benchmarks, especially those stemming from previous work.

7. Please add a discussion section to discuss whether other similar studies are consistent with the results of this study and, if so, to analyze the causes.

8. The writing of the paper needs a lot of improvement in terms of grammar, spelling, and presentations. The paper needs careful English polishing since there are many typos and poorly written sentences.

Minor comments:

1. In the abstract part of the manuscript, the research results should be clearer.

2. Section 3 and section 4 are somewhat related to each other. For conciseness, it is better to combine them together.

3. The authors should recheck the formula numbers. The numbering of the formulas is discontinuous, particularly in Section 4.2. I did not find formulas (6), (9) and (11) in the manuscript. It is highly recommended that the authors proofread the paper carefully for clarity.

4. There are no Supporting Information files in the submitted manuscript. Please add data sources to ensure data availability.

Reviewer #2: You can add more latest references to defend your research, as latest research has been done on this topic, so providing more information will be useful for the readers, you can also quote the new research under way on the related research by other scholars

6. PLOS authors have the option to publish the peer review history of their article (what does this mean?). If published, this will include your full peer review and any attached files.

Reviewer #1: No

Reviewer #2: No

---

## [Author Response · Author response to Decision Letter 0]

7 Mar 2023

Response to Reviewers

Dear Editor and Reviewers:

Thank you for your letter and comments concerning our manuscript entitled “Fresh keeping decision and coordination of fresh agricultural products supply chain under carbon cap-and-trade” (PONE-D-22-33342). Those comments are all valuable and very helpful for revising and improving our paper. We have studied comments carefully and have made correction which we hope meet with approval. Revised portion are marked red in the paper. The main corrections in the paper and the responds to the comments are as follows.

About additional requirements:

Comment 1: Please ensure that your manuscript meets PLOS ONE’s style requirements, including those for file naming.

Response to comment 1: We read the PLOS ONE style templates and made corresponding changes. Including the font of the title of the article, the author’s name, department, unit, nationality, address, and the identity and email address of the author of the communication. See the first page of the article for the details of the revision.

Comment 2: Please remove any funding-related text from the manuscript and let us know how you would like to update your Funding Statement.

Response to comment 2: We have deleted the text related to fundings from the manuscript, and we hope to write the Funding Statement as follows: 

Fundings: The authors gratefully acknowledge financial support from National Natural Science Foundation of China (Grant No.72103178), Social Science Foundation of Jiangsu Province (Grant No.20GLA002), Graduate Research and Innovation Projects of Jiangsu Province (Grant No. SJKY19_2519).

Comment 3: In your Data Availability statement, you have not specified where the minimal data set underlying the results described in your manuscript can be found.

Response to comment 3: All data used in the article has been shown in the first paragraph of Section 4 of the article. According to the reviewer’s comment, we added the Supporting Information file.

Comment 4: PLOS requires an ORCID iD for the corresponding author in Editorial Manager on papers submitted after December 6th, 2016. Please ensure that you have an ORCID iD and that it is validated in Editorial Manager.

Response to comment 4: We have verified the corresponding author’s ORCID iD in Editorial Manager.

About comments given by Reviewer #1:

Major comments：

Comment 1: The inspiration and significance of the manuscript should be added after the abstract.

Response to comment 1: According to the reviewer’s suggestion, we added the inspiration and significance of the manuscript after the abstract. The specific changes are as follows: These conclusions are of great significance to the operation and management of fresh agricultural products suppliers, the improvement of consumers’ quality of life and the protection of ecological environment under carbon cap-and-trade. (The last sentence of the Abstract).

Comment 2: In the Introduction, 1st paragraph must support a practical example of which support the problem descriptions. A potential suggestion to the author/s is to support the importance of a title with some practical examples. However, it seems to be so vague that it lacks credibility.

Response to comment 2: According to the reviewer’s suggestion, we added some examples of implementing carbon cap-and-trade policy within the scope of fresh agricultural products suppliers to prove the necessity of studying the fresh-keeping decision of fresh agricultural products suppliers under carbon cap-and-trade. See the first paragraph of Section 1 for details.

Comment 3: In the Introduction, you need to connect the state of the art to your paper goals. Please follow the literature review by a clear and concise state of the art analysis. This should clearly show the knowledge gaps identified and link them to your paper goals. Please reason both the novelty and the relevance of your paper goals. Clearly discuss what the previous studies that you are referring to. What are the Research Gaps/Contributions? Please note that the paper may not be considered further without a clear research gap and novelty of the study.

Response to comment 3: According to the reviewer’s suggestion, we analyzed the research status after the literature review, and pointed out the gaps in the existing research. And our research just fills these gaps, so it has certain novelty. See paragraph 5 of section 1 for specific changes.

Comment 4: The paper does not provide any background information about the carbon cap-and-trade policy. Hence, it makes the paper not very coherent, and difficult to follow. Please give a specific explanation of the technical term “the carbon cap-and-trade policy” in the manuscript. Additionally, the link between policy implementation and fresh produce preservation should be discussed in depth.

Response to comment 4: According to the reviewer's suggestion, we explained the meaning of carbon cap-and-trade policy, and discussed the relationship between the implementation of carbon cap-and-trade policy and the fresh-keeping efforts of suppliers (Paragraph 1 of Section 1). Moreover, on the basis of the relationship analysis, we put forward the research questions of this paper (Paragraph 2 of Section 1).

The explanation of carbon cap-and-trade policy is as follows: Carbon cap-and-trade means that the government decomposes the total amount of carbon emission rights into a certain unit of carbon emission rights, and then allocates the emission rights to carbon dioxide emission source enterprises in a specific way, and allows them to buy and sell carbon emission rights in the market.

The relationship between the implementation of carbon cap-and-trade policy and the fresh-keeping efforts of suppliers: Fresh-keeping is not only an important part of the operation of fresh agricultural products supply enterprises, but also the main source of carbon emissions of fresh agricultural products supply enterprises. Therefore, the full implementation of the carbon cap-and-trade policy has brought a difficult problem to the suppliers of fresh agricultural products: improving fresh-keeping efforts will bring more carbon emission costs, while reducing fresh-keeping efforts can reduce carbon emission costs but cannot meet consumers’ demand for freshness.

Comment 5: In Example Analysis section, all the figures are so vague that it is impossible to distinguish the exact meaning after reading them. Please upload figures in strict accordance with the journal format.

Response to comment 5: We uploaded figures in strict accordance with the journal format.

Comment 6: The conclusion is very weak. Please make sure your conclusions' section underscores the scientific value-added of your paper, and/or the applicability of your findings/results. Highlight the novelty of your study. In addition to summarizing the actions taken and results, please strengthen the explanation of their significance. It is recommended to use quantitative reasoning comparing with appropriate benchmarks, especially those stemming from previous work.

Response to comment 6: We rewrote the Conclusion. On the one hand, we summarized the work of this paper. On the other hand, we compared the optimal fresh-keeping decisions and benefits under different parameter conditions, and analyzed the practical significance and reasons of different results. For the more important and novel conclusions, such as points (1), (2) and (5), we explained them in detail. See section 5 for specific modifications.

Comment 7: Please add a discussion section to discuss whether other similar studies are consistent with the results of this study and, if so, to analyze the causes.

Response to comment 7: We added a discussion part to discuss the similarities and differences between the research of Ma et al. and this paper, and further explained the research significance of this paper. See paragraph 7 of Section 5 for details.

Comment 8: The writing of the paper needs a lot of improvement in terms of grammar, spelling, and presentations. The paper needs careful English polishing since there are many typos and poorly written sentences.

Response to comment 8: We have revised some typos and inappropriate sentences in the article. At the same time, we also invited native English speakers to polish our article. The specific changes in the text have been marked in red.

Minor comments:

Comment 1: In the abstract part of the manuscript, the research results should be clearer.

Response to comment 1: We rewrote the abstract and introduced the research results of this paper more clearly. See Abstract of the paper.

Comment 2: Section 3 and section 4 are somewhat related to each other. For conciseness, it is better to combine them together.

Response to comment 2: According to the suggestion, we put Section 3 and Section 4 together. See Section 3 of the paper.

Comment 3: The authors should recheck the formula numbers. The numbering of the formulas is discontinuous, particularly in Section 4.2. I did not find formulas (6), (9) and (11) in the manuscript. It is highly recommended that the authors proofread the paper carefully for clarity.

Response to comment 3: We renumbered the formulas (6)-(8).

Comment 4: There are no Supporting Information files in the submitted manuscript. Please add data sources to ensure data availability.

Response to comment 4: All data used in the article has been shown in the first paragraph of Section 4 of the article. And according to the reviewer’s comment, we added the Supporting Information file.

About comments given by Reviewer #2: 

Comment: You can add more latest references to defend your research, as latest research has been done on this topic, so providing more information will be useful for the readers, you can also quote the new research under way on the related research by other scholars

Response to the comment: According to the suggestion, we added some newer references to defend our research. Including reference [5], [6], [7], [9], [10], [14], [18], [19]. 

Specifically:

[5] Liu YP, Yan B and Chen XX. Coordination of dual-channel supply chains with uncertain demand information. IMA Journal of Management Mathematics, 2022, DOI: 10.1093/imaman/dpac011

[6] Bai SZ, Lv Y and Liu ZJ. Optimal decision and coordination of fresh e-commerce supply chain considering double loss. Discrete Dynamics in Nature and Society, 2022, DOI: 10.1155/2022/3781698

[7] Hu HJ, Li YK, Li MD, et al. Optimal decision-making of green agricultural product supply chain with fairness concerns. Journal of Industrial and Management Optimization, 2022, DOI: 10.3934/jimo.2022155

[9] Zhang QY, Cao W and Zhang ZC. Operational decisions and game analysis in the agricultural supply chain: invest or not? Kybernetes, 2021, DOI: 10.1108/K-07-2021-0585

[10] Yan B, Chen YR and He SY. Decision making and coordination of fresh agriculture product supply chain considering fairness concerns. Rairo-Operations Research, 2019, DOI: 10.1051/ro/2019031

[14] Luo M, Zhou GH and Xu H. Three-tier supply chain on temperature control for fresh agricultural products using new differential game model under two decision-making situations. Operations Management Research, 2022, 15(3-4): 1028-1047.

[18] Yang L, Ji JN, Wang MZ and Wang ZZ. The manufacturer’s joint decisions of channel selections and carbon emission reductions under the cap-and-trade regulation. Journal of Cleaner Production, 2018, 193: 506-523.

[19] Zhang DM, Shi GH, Yao GX, et al. Decisions of ordering and preservation for fresh agricultural products under carbon emission constraints. Industrial Engineering and Management, 2020, 25: 145-151.

We tried our best to improve the manuscript and made some changes in the manuscript. These changes will not influence the content and framework of the paper. 

We appreciate for Editors and Reviewers’ warm work earnestly, and hope that the correction will meet with approval. Once again, thank you very much for your comments and suggestions!

Finally, I sincerely hope to get a reply as soon as possible, because I am about to graduate and need a paper to meet the requirements of graduation. Thank you very much!

Best Regards,

YangYang

---

## [Decision Letter · Decision Letter 1]

20 Mar 2023

Fresh keeping decision and coordination of fresh agricultural products supply chain under carbon cap-and-trade

PONE-D-22-33342R1

Dear Dr. Guanxin Yao,

We’re pleased to inform you that your manuscript has been judged scientifically suitable for publication and will be formally accepted for publication once it meets all outstanding technical requirements.

Kind regards,

Bo Huang

Academic Editor

PLOS ONE

Additional Editor Comments (optional):

Reviewers' comments:

Reviewer's Responses to Questions

**Comments to the Author**

1. If the authors have adequately addressed your comments raised in a previous round of review and you feel that this manuscript is now acceptable for publication, you may indicate that here to bypass the “Comments to the Author” section, enter your conflict of interest statement in the “Confidential to Editor” section, and submit your "Accept" recommendation.

Reviewer #1: All comments have been addressed

2. Is the manuscript technically sound, and do the data support the conclusions?

Reviewer #1: Yes

3. Has the statistical analysis been performed appropriately and rigorously? 

Reviewer #1: Yes

4. Have the authors made all data underlying the findings in their manuscript fully available?

Reviewer #1: Yes

5. Is the manuscript presented in an intelligible fashion and written in standard English?

Reviewer #1: Yes

6. Review Comments to the Author

Reviewer #1: The author carefully revised their manuscript. Fortunately, the quality of this manuscript has improved significantly.

7. PLOS authors have the option to publish the peer review history of their article (what does this mean?). If published, this will include your full peer review and any attached files.

Reviewer #1: No

---

## [Editor Report · Acceptance letter]

24 Mar 2023

PONE-D-22-33342R1 

Fresh keeping decision and coordination of fresh agricultural products supply chain under carbon cap-and-trade 

Dear Dr. Yao:

I'm pleased to inform you that your manuscript has been deemed suitable for publication in PLOS ONE. Congratulations! Your manuscript is now with our production department. 

Kind regards, 

on behalf of

Professor Bo Huang 

Academic Editor

PLOS ONE